# A Bibliometric and Word Cloud Analysis on the Role of the Internet of Things in Agricultural Plant Disease Detection

Rutuja Rajendra Patil [1], Sumit Kumar [1,*], Ruchi Rani [2], Poorva Agrawal [3] and Sanjeev Kumar Pippal [4]

1 Symbiosis Institute of Technology, Pune, Symbiosis International (Deemed University), Pune 412115, Maharashtra, India
2 Department of Computer Science, Indian Institute of Information Technology, Kottayam 686635, Kerala, India
3 Symbiosis Institute of Technology, Nagpur, Symbiosis International (Deemed University), Pune 440008, Maharashtra, India
4 Department of Technology, NSBT, MGM University, Aurangabad 431005, Maharashtra, India
* Correspondence: er.sumitkumar21@gmail.com

**Abstract:** Agriculture has observed significant advancements since smart farming technology has been introduced. The Green Movement played an essential role in the evolution of farming methods. The use of smart farming is accelerating at an unprecedented rate because it benefits both farmers and consumers by enabling more effective crop budgeting. The Smart Agriculture domain uses the Internet of Things, which helps farmers to monitor irrigation management, estimate crop yields, and manage plant diseases. Additionally, farmers can learn about environmental trends and, as a result, which crops to cultivate and how to apply fungicides and insecticides. This research article uses the primary and subsidiary keywords related to smart agriculture to query the Scopus database. The query returned 146 research articles related to the keywords inputted, and an analysis of 146 scientific publications, including journal articles, book chapters, and patents, was conducted. Node XL, Gephi, and VOSviewer are open-source tools for visualizing and exploring bibliometric networks. New facets of the data are revealed, facilitating intuitive exploration. The survey includes a bibliometric analysis as well as a word cloud analysis. This analysis focuses on publication types and publication regions, geographical locations, documents by year, subject area, association, and authorship. The research field of IoT in agricultural plant disease detection articles is found to frequently employ English as the language of publication.

**Keywords:** bibliometric analysis; plant disease detection; word cloud; Internet of Things (IoT)

## 1. Introduction

The need to increase farm production has become crucial due to diminishing agricultural fields and the loss of scarce natural resources [1,2]. The lack of natural resources such as fresh water and arable land, as well as decreased yield rates in many staple commodities, have made the issue worse [3,4]. Another looming issue for the farming sector is the shifting demographics of the agricultural workforce. Internet of Things strategies are targeted at assisting farmers in bridging the supply-demand gap by maintaining high yields, sustainability, and environmental security [5]. Increasing crop yields while reducing operating costs can be accomplished through precision agriculture, which uses IoT technologies [6,7]. Agricultural IoT innovations provide advanced devices and wireless networking [8,9]. Based on IoT technology, smart farming assists growers and producers in reducing pollution and increasing efficiency in a variety of ways, including the quantity of fertilizer used, the number of trips made by farm cars, and the efficient use of resources such as water and power [10–12]. IoT smart farming solutions is a device that uses sensors to track the crop field and automate the irrigation system (light, humidity, temperature, soil moisture, crop protection, etc.) [13–15]. The agriculture sector is in its infancy when technological aspects are considered. In traditional farming, farmers have to bear financial

losses due to various reasons such as incorrect weather prediction, amount of pesticides and fertilizers to be sprayed, incorrect irrigation, plant diseases and pests, and low yield of crops [16,17]. The younger generation of farmers reining in the agricultural business now demands a smart system for improving the production of crops [18,19]. The issues of traditional farming and the demands of younger farmers will be addressed by introducing IoT in the agricultural sector with cost-effective solutions. Crop disease forecasting is crucial because it helps farmers plan their crop production and minimise risks. Using information gathered from farm sensors, artificial networks are utilized to forecast crop illnesses and pests [20,21]. This information includes variables including soil, temperature, pressure, rainfall, and humidity [22–25]. This information also helps in estimating the severity of the crop diseases [26].

Bibliometric Analysis of the role of IoT in Smart Agriculture: The base word, bibliometric, is made up of two words that come from Greek and Latin words, where 'biblio' means book or paper and 'metrics' means to measure. A.J. Pritchard introduced the concept of bibliometrics in 1969. In accordance with his definition, bibliometrics refers to the application of mathematical and statistical approaches and models to books and some other forms of written communication, including journals, e-books, theses, reports, etc. Once the article is published, it is necessary to understand its influence and its impact [27,28]. The impact can be understood by the number of citations received to a particular research paper. Thus, to understand the influence and impact of the research document, it is necessary to conduct a bibliometric analysis. Lotka's Law, Bradford's Law, and Zipf's Law are the three bibliometrics laws [29].

1.  Lotka's law: It was introduced in 1934 and is also known as the author's productivity. It measures the productivity of an author based on the number of published articles.
2.  Bradford's law: It is known as the law of scattering. It measures the scattering of articles on a particular subject in various periodicals.
3.  Zipf's Law: It is known as the law of word frequency, and is used to predict the occurrence of the words in the article.

Thus, a bibliometric analysis is useful for researchers who are beginners. This analysis provides insights into what kind of research work is already conducted by the researcher in the field. Additionally, it can be used to identify the areas of research that need further study.

There is a need to carry out a bibliometric survey of plant leaf diseases to gain a better understanding of this research area. In this bibliometric study, the following goals are pursued:

- Identifying different categories of publications in the domain of IoT in Smart Agriculture.
- Discovering different types of language used for research publications.
- Analyzing the year-wise trends of publishing the research articles.
- Identifying geographical locations contributing more to IoT in the Smart Agriculture domain;
- Analyzing trends by source type.
- Identifying majorly contributing authors in the Smart Agriculture domain.
- Determining trends in publications based on affiliations.
- Conducting a citation analysis of the publications.

For investigating and analysing huge quantities of scientific data, a bibliometric analysis is an established and efficient technique. It allows one to explore the peculiarities of a certain field's evolutionary development while highlighting its horizons. However, its use in business research is still somewhat new and frequently underdeveloped. As a result, to provide an overview of the bibliometric approach, with a special emphasis on its various methodologies, as well as step-by-step directions that can be trusted to undertake the bibliometric analysis rigorously and with confidence. Lastly, it also emphasizes on when and how to apply a bibliometric analysis in comparison to other related strategies such as systematic literature reviews and meta-analysis.

This research work presents a bibliometric analysis of the role of IoT in smart agriculture in Section 2. This section highlights an initial collection of data related to the IoT in smart agriculture. An analysis of the data from Scopus that were extracted is shown in Section 3. This part presents two different sorts of analyses: the first is network analysis, and the second is statistical analysis. The top three cited research publications are represented in a word cloud in Section 4. The conclusions made from the research are described in Section 5. The paper ends with a list of references.

## 2. Foundational Data Acquisition

There are two ways to access research-based articles, book chapters, documents, and publications: open access and paid access. Paid access is when the research documents can be accessed by paying the fees to access the respective document. Open access is when the research documents can be accessed without paying any charges to access the document. The publications in open access can be simply accessed by completing a registration process at the corresponding website. The other way to access the articles is to access them via the library portals of the institutes. A supplementary method to access the research data is to obtain access to research databases. Scopus, Web of Science, Science Direct, Research Gate and Google Scholar, etc., are a few of the popular databases which can be accessed [30]. This research paper considers one of the most considered research databases, named the Scopus database. This comprehensive research database has highly relevant research documents such as articles, journals, conference proceedings, and book chapters. The database articles fall under various research disciplines, namely engineering, science, management, arts, medical, and agriculture [31]. The research articles submitted to the journals are reviewed by a scientific process called a peer review. These manuscripts are critically reviewed by the experts in that field and provide feedback on whether the manuscript can be accepted, rejected, or needs revisions. The database spans international research articles. This is why Scopus is regarded as a precise inception point for research [32]. This manuscript considers the Scopus dataset in alliance with distinguished keywords identified in Section 2.1.

### 2.1. Distinguish Keywords

The distinguish keywords that correspond to IoT with agriculture are classified into two segments, namely: principal and subsidiary keywords. Table 1. exhibits the choice of search keywords that are utilized as a searching strategy for this research. Therefore, the following is the search query that is used to find the documents in a Scopus dataset: "IOT" AND "Sensors" AND "agriculture" AND "climate" AND "prediction" AND "Artificial Intelligence" AND "crop diseases" OR "diseases" OR "temperature" OR "humidity" OR "moisture" OR "weather" OR "machine learning" OR "deep learning".

**Table 1.** Selection of search keywords for IoT in agriculture.

| Principal Keyword | "IOT" |
|---|---|
| Subsidiary keyword using (AN) | "Sensors" AND "agriculture" AND "climate" AND "prediction" AND "Artificial Intelligence" AND "crop diseases" |
| Subsidiary keywords using (OR) | "diseases" OR "temperature" OR "humidity" OR "moisture" OR "weather" OR "machine learning" OR "deep learning" |

### 2.2. Preparatory Search Result

The primitive unit of this research manuscript is the Scopus dataset. In the first instance, a query was executed on the Scopus database, which included both principal and subsidiary key words as a search strategy and had an outcome of 146 publications. The outcome of the query included 146 publications that are published as well as unpublished. Table 2 depicts the classification of various types of publications in the research area of IoT

in agriculture. The researchers majorly preferred articles in which to publish their work with a 51.37% contribution, and this was followed by reviews, which contributed 24.65%. It can be clearly perceived that the researchers least preferred books in which to publish their publications.

**Table 2.** Classification of Publications for IOT in agriculture.

| Category of Publications | Publication Count | Percentage |
| --- | --- | --- |
| Article | 75 | 51.37 |
| Review | 36 | 24.65 |
| Conference Paper | 22 | 15.07 |
| Book Chapter | 8 | 5.49 |
| Book | 5 | 3.42 |

The outcome documents are analyzed based on different types of language used for publishing the papers. Table 3 represents various languages used for publishing documents on IoT in agriculture. The researchers demonstrated their inclination towards the English language to publish their publications, while other languages, such as Chinese, are used rarely by the researchers to publish the articles.

**Table 3.** Languages used for publishing on IOT in crop disease detection.

| Publication Language | Publication Count |
| --- | --- |
| English | 145 |
| Chinese | 1 |

*2.3. Highlights in Exploratory Data*

From 2013 to 2020, records related to IoT in crop disease detection were retrieved over a seven-year period. Table 4 demonstrates annual developments in the number of publications in the field of the Internet of Things in crop disease identification. It can be quickly deduced from these data that the study field contributed most in the year 2020 for publishing research documents with a total of 74 documents released. However, between 2013 and 2016, only a few studies were performed. In the year 2014, there are no publications.

**Table 4.** Yearly publishing trends of IOT in crop disease detection.

| Year | Publication Count |
| --- | --- |
| 2020 | 74 |
| 2019 | 38 |
| 2018 | 14 |
| 2017 | 10 |
| 2016 | 3 |
| 2015 | 4 |
| 2014 | 0 |
| 2013 | 1 |

*2.4. Data Interpretation*

A thorough bibliometric analysis is conducted in Section 3 to understand the diversity of the extracted material, categorize pertinent researchers, and examine issue statements in the area of IoT in agricultural disease detection. The study locations, the contributors' connections, the authors' names, and the journals where the research articles were published all show distinctiveness in the research field. The search analysis is carried out using the keywords found in the extracted material. The recovered information is frequently used to estimate the number of citations for each research paper and collaborative study.

Using a line chart to schematically portray Table 4, Figure 1 demonstrates that 2020, with a total of 74 documents published, is the most frequently occurring year for research document publication.

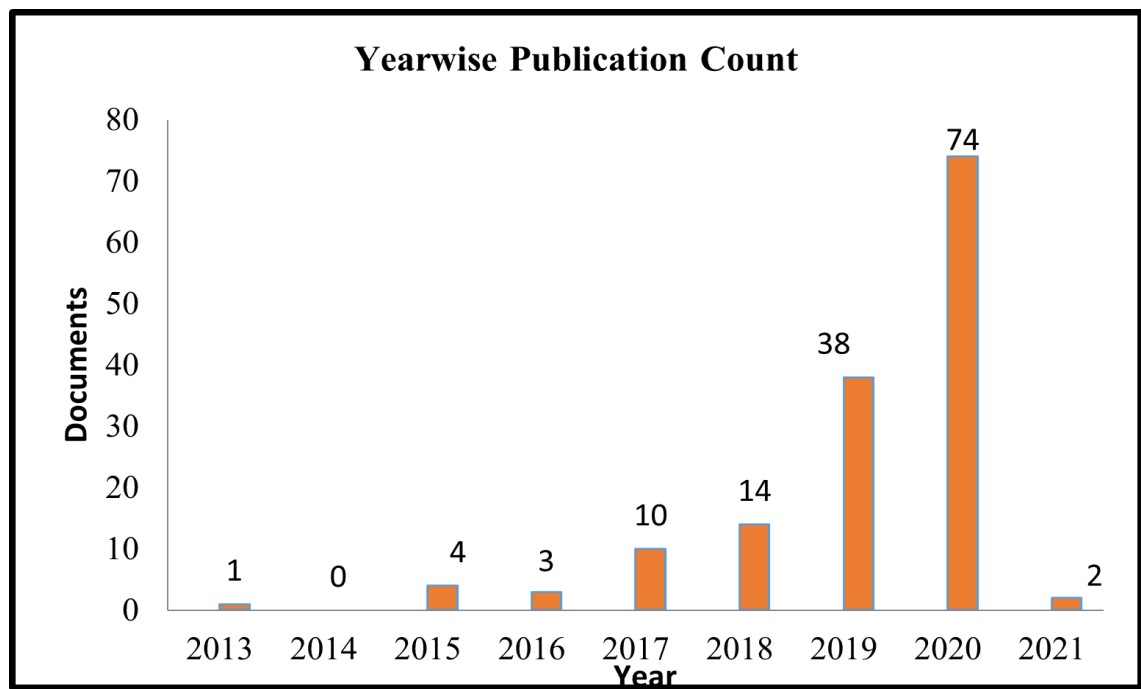

**Figure 1.** Yearly publishing trends of IoT in crop disease detection.

## 3. Biblometric Review

A bibliometric survey on the application of IoT in crop disease detection is conducted using two alternative methods.

1.  Statistical analysis focuses on author affiliations, authors, source types, source names, and contributions to the study field by region and subject area.
2.  A review of networked data that concentrated on geographical regions, keywords, source names, publication titles, citation counts, and collaborative research with other research scholars.

### 3.1. Spatial Location-Based Analysis

The information on academic articles from various nations is depicted in Figure 2. This interactive map was made with Google Sheets in a collaborative environment [33]. A Google spreadsheet receives data from an excel document with two columns for the nation and number of publications, then uses this information to create a map. The publication count information for each country on this global map can be read using a scale. Smaller contributions to the research area correspond to smaller dots on the map. At the left-hand bottom of the page is a scale representing the number of publications. The United States is indicated by a dark green dot on the map, indicating that it has the most articles published (26 total) on the use of IoT to identify plant diseases.

Figure 3 illustrates the top ten countries with publications in the field of smart agriculture. The bar graph reveals that the United States has the largest share of publicizing documents in the smart agriculture research sector, with 26 contributions, followed by India with 19 contributions. France, Malaysia, and Pakistan are the countries that contribute the least to research in the field of smart agriculture.

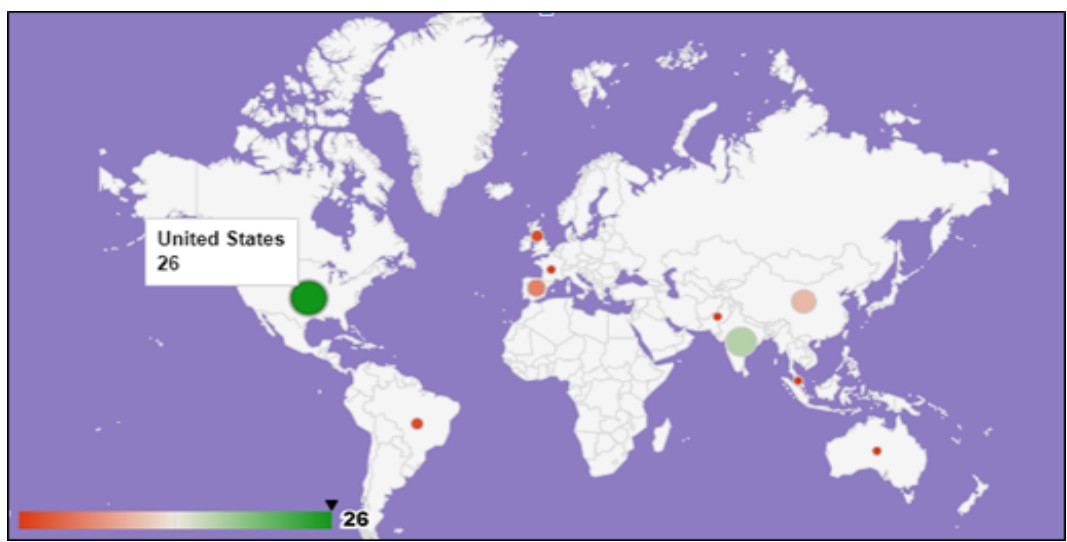

**Figure 2.** Geographical span of research on plant disease detection using IoT.

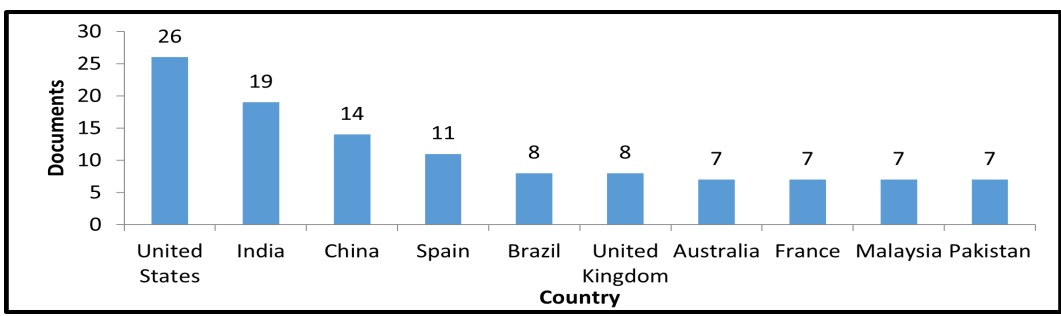

**Figure 3.** Top ten countries for IoT in plant disease detection research publications.

*3.2. Statistical Analysis Based on Keywords*

Table 5 displays the top ten keywords for smart agriculture that were checked in the Scopus database. The right mix of keywords will help to find publications in the field for which researchers are searching. It offers a general description for what the researchers are searching. Since the Internet of Things is the most frequently used keyword in the extracted literature, it is clear that the majority of research in the field of plant disease detection is focused on it.

**Table 5.** Top ten keywords for IoT in plant disease detection.

| Classification of Publications | Number of Publications |
|---|---|
| Internet of Things | 64 |
| Artificial Intelligence | 29 |
| Internet of Things (IoT) | 26 |
| Precision Agriculture | 25 |
| Agriculture | 23 |
| Forecasting | 22 |
| Wireless Sensor Networks | 20 |
| Agricultural Robots | 19 |
| Crops | 19 |
| IoT | 19 |

*3.3. Network Analysis*

An analysis of networks is a graphical representation of interactions between different computational attributes. For the same reason, there are a variety of resources available. The graphs used in this research paper were created using VOSviewer, Gephi, and NodeXL.

Figures 4–9 shows network analysis diagrams for a combination of various computeable parameters in the field of IoT in plant disease detection based on literature from the Scopus database. The VOSviewer tool can be downloaded for free. Based on the parameters, the tool evaluates the bibliometric network. A file with a .csv extension from Scopus is loaded into VOSviewer as the input [34,35]. Three forms of visualization analysis are available: networks, overlays, and densities. Figure 4 displays a network visualization chart based on a mix of Scopus keywords and source names. The keywords used in the source titles of extracted documents are depicted in circles on the map. The keyword is used more frequently as the circle gets larger. The connections between the circles indicate how far apart two keywords are from one another. The relation size decreases in proportion to the strength of the association between the terms. Similar-colored keywords describe groups of keywords that are closely connected. There are six clusters in the diagram, and each one is represented by a distinct hue.

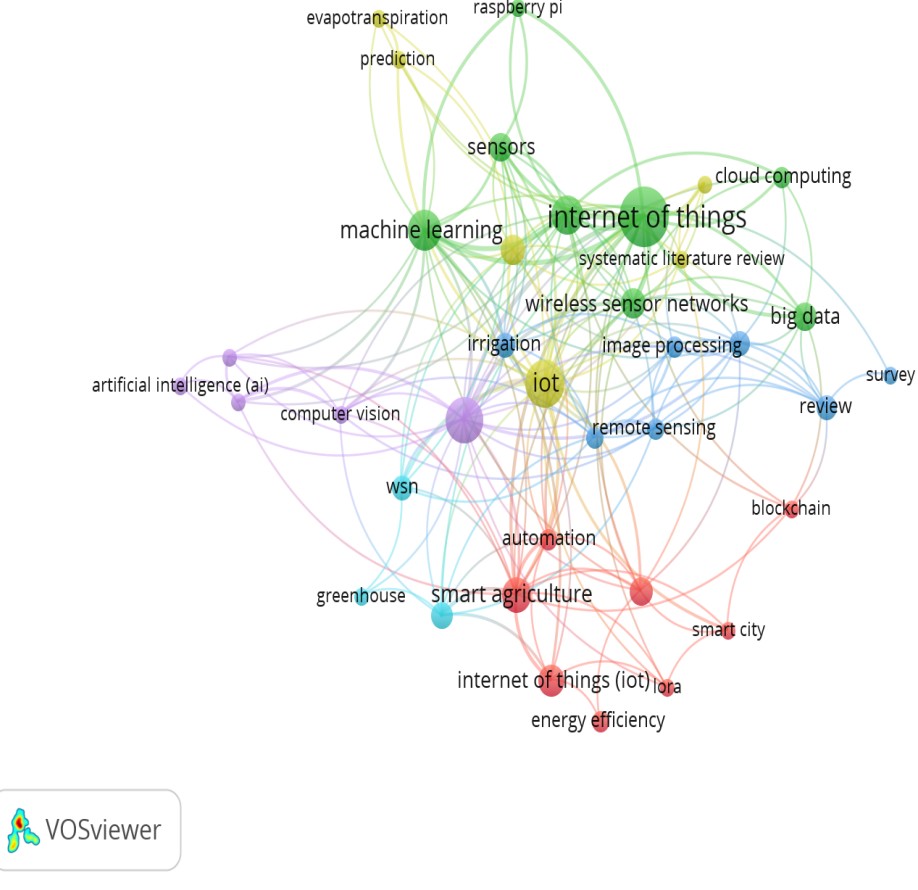

**Figure 4.** Source title and keywords-based network visualization diagram.

Figure 5 illustrates a cluster of publication titles and their corresponding publication years. This diagram was created using the NodeXL open source network research diagramming tool. Different bodies, such as the publication title and publication year, are represented by nodes. Edges are the relations that occur between these individuals. The data from the publication year and title are traced using the Fruchterman-Reingold arrangement. The cluster size suggests that the bulk of plant leaf disease publications was released in 2020, followed by 2019. There are no publications in the year 2014.

Figure 6 illustrates a group of co-writers and authors who feature in several articles. The writer's joint contributions are clear. The relation reflects the authors' joint work on the documents that have been written. The threshold value for authors with a minimum of two publications was manually set to two, resulting in 59 authors. Figure 6 estimates and

shows the total strength of the co-authorship links with other authors. The network chart of journal titles and the citations obtained from publications published in it is represented in Figure 7. This diagram was created using Gephi, an open source program. To plot the diagram, the Fruchterman-Reingold layout is used. The publication title is a joint work of the contributors, so there are 142 nodes and 234 edges in the style. An in degree property was set on the edges, indicating that arrows heading to a certain node have referenced the article. The journal title with the most citations is represented by the dark green color dot.The color gradient is based on the number of citations to a particular article.

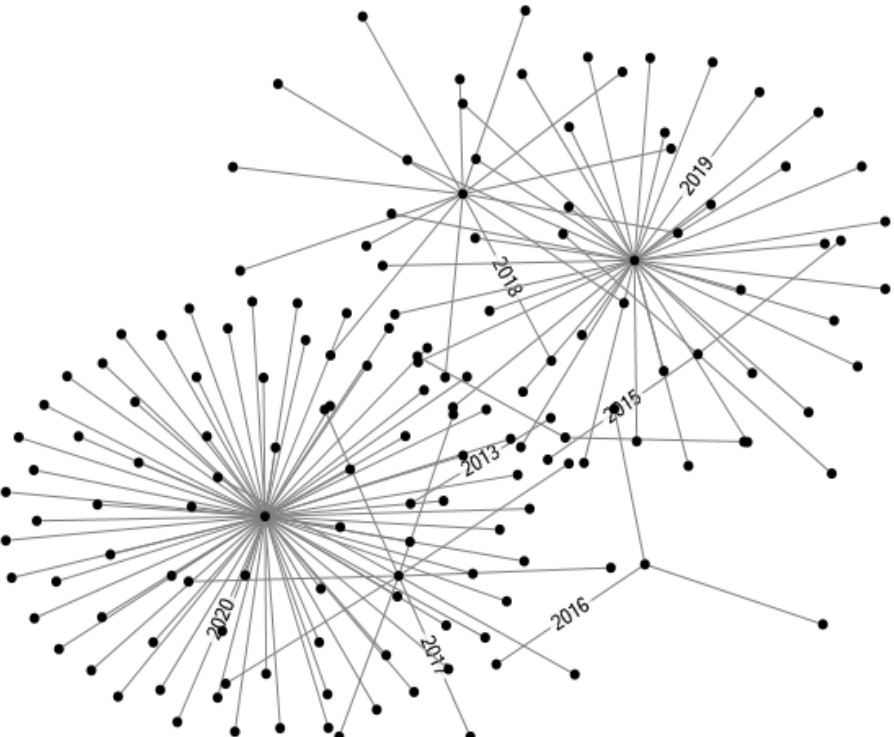

**Figure 5.** Network analysis showing the cluster of publishing year and title.

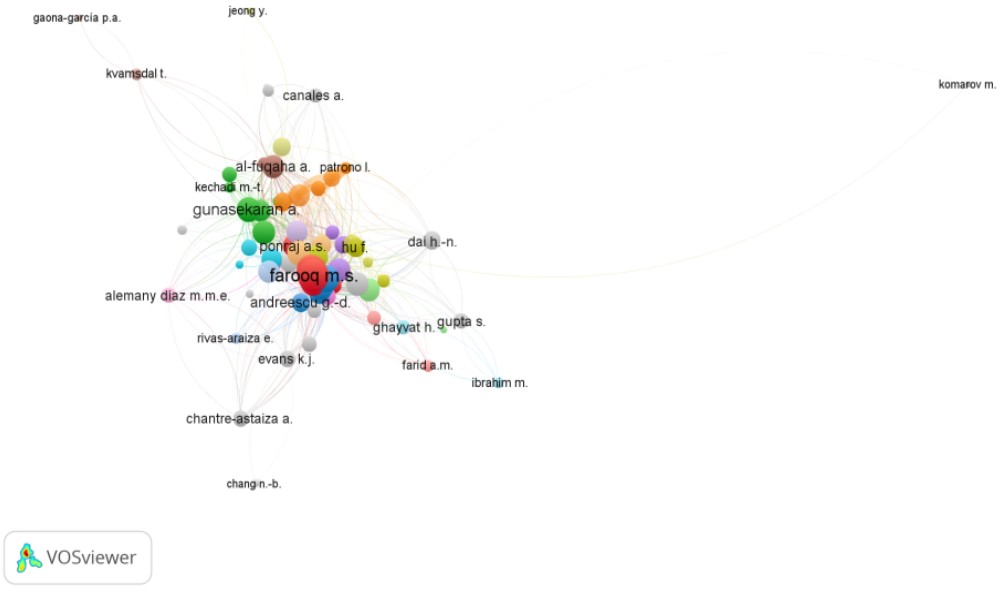

**Figure 6.** Network analysis of co-authors and authors based on co-appearance among the same papers.

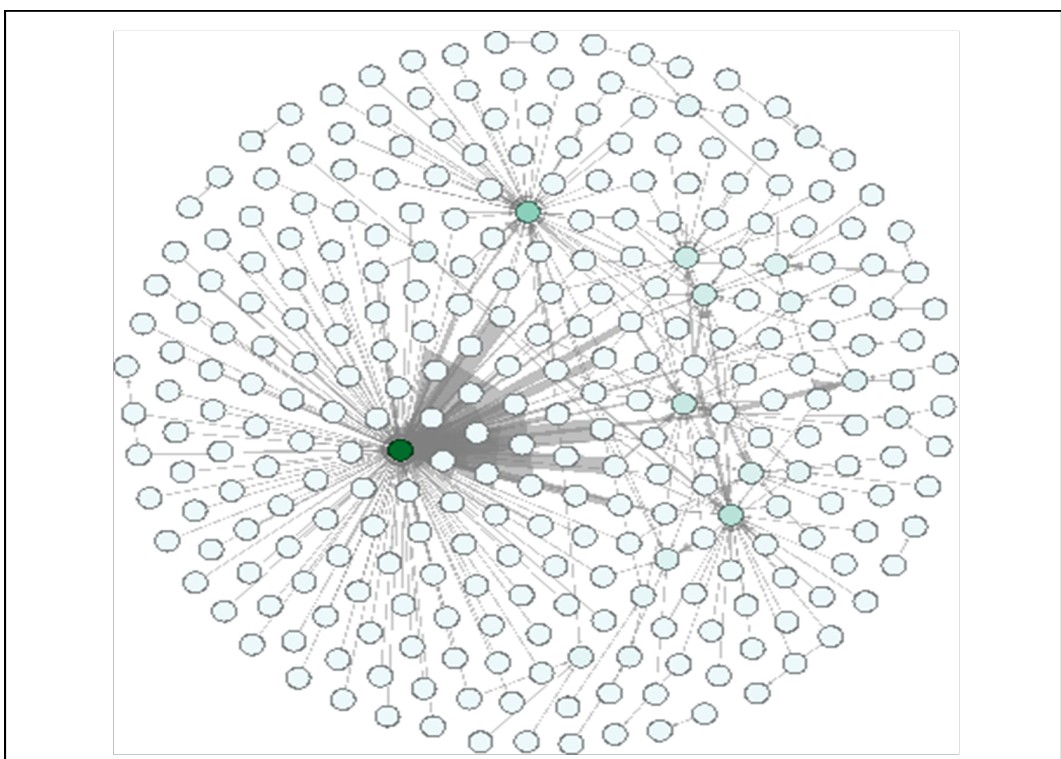

**Figure 7.** Publication title network map and citations received by publications (Source: https://gephi.org/ (accessed on 21 December 2020)).

*3.4. Statistical Analysis Pertaining to Subject Areas*

Figure 8 categorizes the publications that were found for IoT in agriculture literature across different fields. This statistic also makes it clear that the most research is conducted in the field of computer science, with 75 publications, followed by the fields of engineering, physics, and astronomy, as well as agricultural and biological sciences.

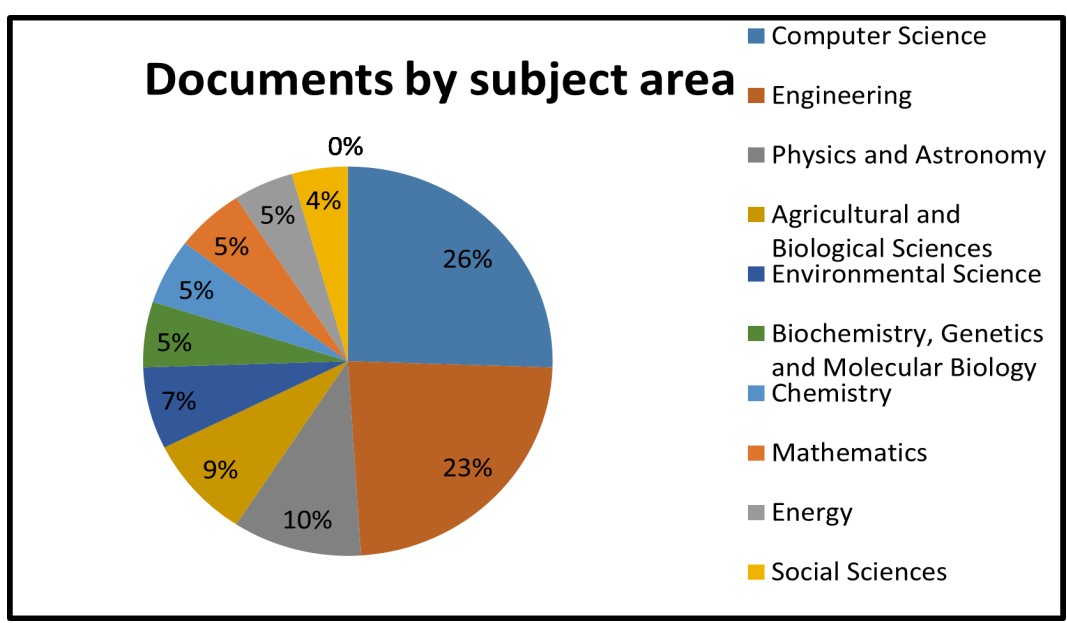

**Figure 8.** Subject area-wise analysis for IoT in agriculture-retrieved literature.

### 3.5. Statistical Analysis Pertaining to Affiliations for IoT in Agriculture

Figure 9 represents the top ten universities and organizational affiliations that contribute the most to the field of IoT in the agriculture domain. The Skolkovo Institute of Science and technology displays the highest contribution towards the research in the field of IoT in agriculture followed by Purdue University. The universities all across the world are showing interest in working in this research area.

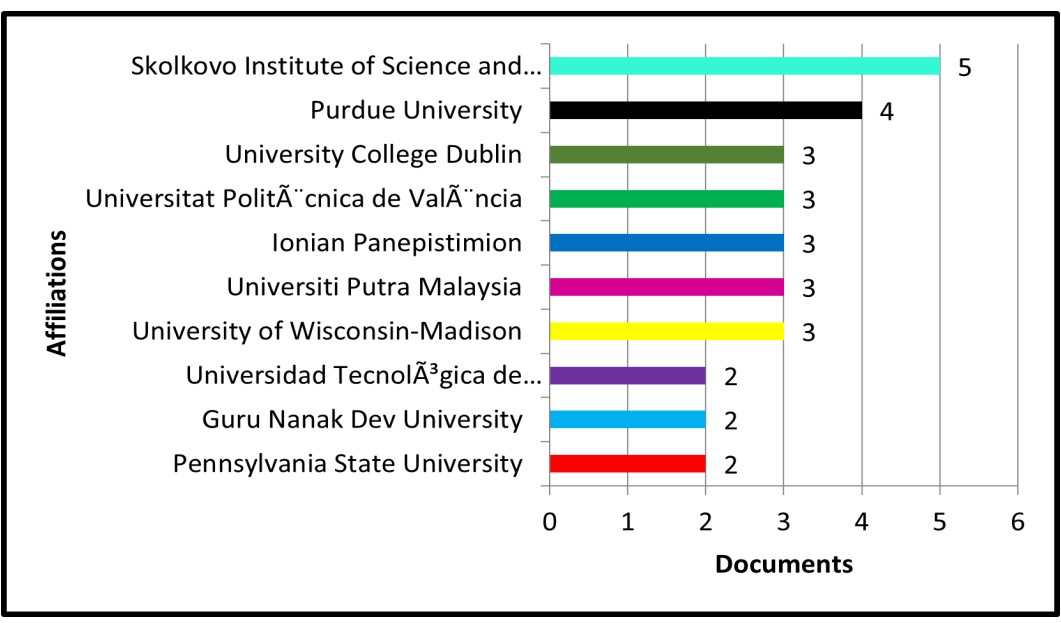

**Figure 9.** Statistics of affiliation for IoT in agriculture.

### 3.6. Statistical Analysis Pertaining to Authors Contributing to IoT in Agriculture

Figure 10 displays the top ten authors contributing in the area of IoT in agriculture. This representation helps to identify the influence of a particular author in a particular field. Authors Dmitrii Shadrin and Andrey Somov, from the Skolkovo Institute of Science and Technology, are the topmost contributors to this field of research, with each having an equal number of publications.

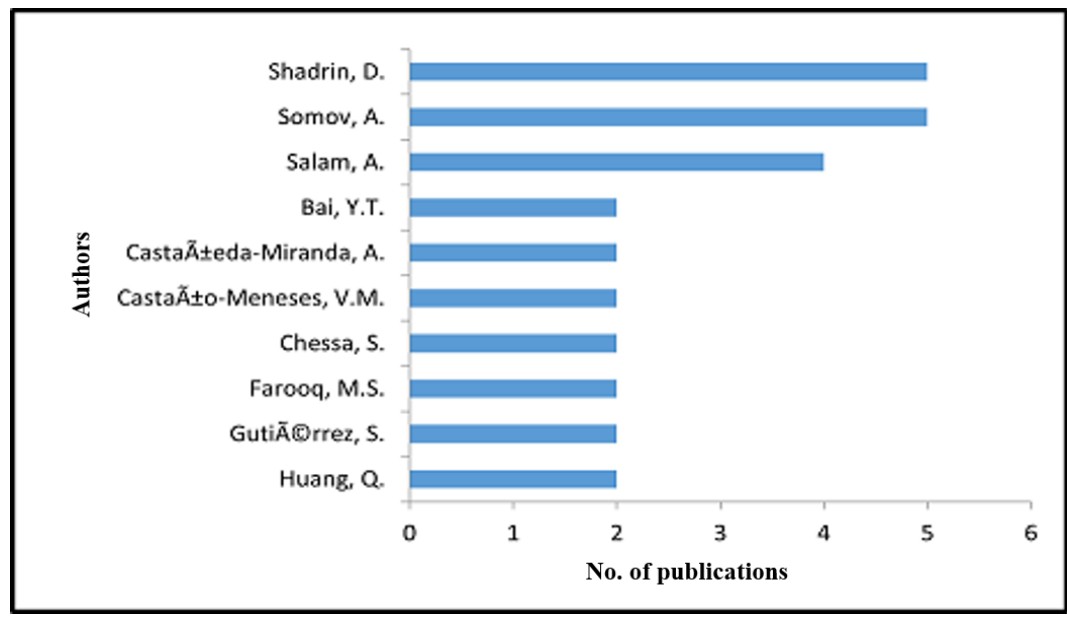

**Figure 10.** Top ten authors contributing to research area of IoT in agriculture.

### 3.7. Statistical Analysis Pertaining to Types of Source

As per Figure 11. the researchers publish their original research work or scholarly articles and manuscripts into a specific type of source. There are various types of sources, namely primary, secondary, and tertiary sources. However, the researchers mainly opt for primary sources to publish their original articles. It can be explicitly stated from the Scopus-extracted IoT in agriculture literature that 51% of the publications are written as articles. Articles are written for the general audience. It is also observed that 25% of publications are reviews and 15% are submitted in conference proceedings.

After reviewing the review papers, no bibliometric review paper was retrieved for the IoT in agriculture. Therefore, this becomes one of the motivations to write this bibliometric review on the IoT in agriculture.

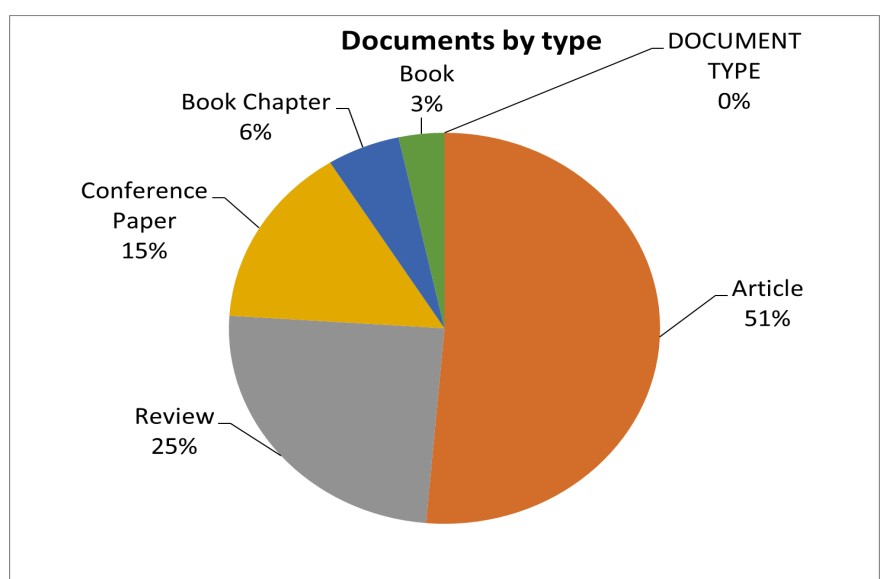

**Figure 11.** Source types of publications for IoT in agriculture.

### 3.8. Statistical Analysis Pertaining to Publication Citations for IoT in Agriculture

Table 6 displays citations obtained from publications extracted in the field of IoT in agriculture on an annual basis. The cumulative citation count of 146 publications is 1489 so far. Very few research papers have been cited from the year 2013 to 2015 with a total count of seven citations, although the highest number of citations is shown in the year 2020 followed by 2019. Figure 12 represents the year-wise statistics of citations acquired by the publications in the field of IoT in agriculture.

By observing Table 6, it can be inferred that there is a total of 1489 citations in publications in the field of IoT in agriculture. The year 2020 is the year that cited the articles the most.

Table 7 represents the top five publication titles derived from the Scopus dataset that have obtained the highest number of citations to date. It is clearly visualized that the research article with the title 'Wireless sensor networks for agriculture: The state-of-the-art in practice and future challenges' has gained the highest number of citations in the research area of IoT in agriculture.

**Table 6.** An analysis based on publication citations for IoT in agriculture.

| Year | No. of Citations |
| --- | --- |
| 2020 | 594 |
| 2019 | 194 |
| 2018 | 104 |
| 2017 | 31 |
| <2016 | 7 |

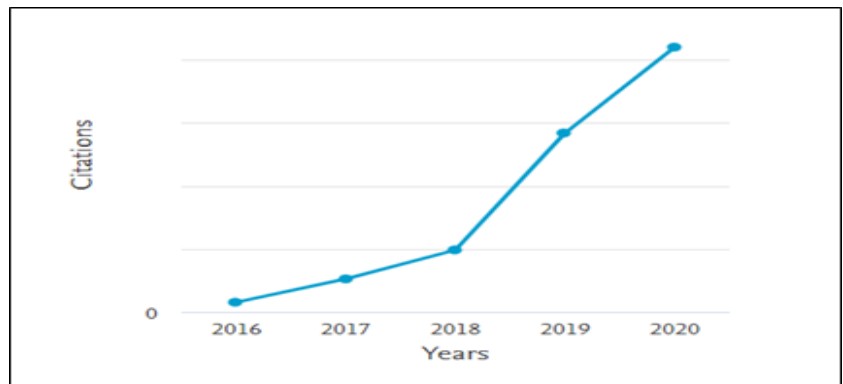

**Figure 12.** Year-wise statistics of publication citations for IoT in agriculture.

**Table 7.** Statistical analysis of top five publications based on citations for IOT in agriculture.

| Document Title (Year) | <2016/2016 | 2017 | 2018 | 2019 | 2020 | Total Citation |
|---|---|---|---|---|---|---|
| Wireless sensor networks for agriculture: The state of the art in practice and future challenges (2015) | 23 | 50 | 71 | 106 | 79 | 330 |
| Deep learning for IoT big data and streaming analytics: A survey (2016) | 0 | 0 | 7 | 84 | 126 | 220 |
| Big Data and cloud computing: innovation opportunities and challenges (2017) | 0 | 20 | 53 | 88 | 55 | 216 |
| Review of IoT applications in agro-industrial and environmental fields (2018) | 0 | 1 | 14 | 57 | 51 | 123 |
| State-of-the-art in artificial neural network applications: A survey (2019) | 0 | 0 | 0 | 35 | 66 | 103 |

### 3.9. Statistical Analysis Pertaining to Source Titles for IoT in Agriculture

The top five source titles retrieved from the literature are analyzed statistically and are shown in Figure 13 for IoT in agriculture publications. It can be concluded that the overall number of publications in the source title named the *Computer and Electronics in Agriculture* is the highest followed by the source title named the Sensors Switzerland. However, other source titles, namely *IEEE Access*, *IEEE Internet of Things Journal* and *Sustainability Switzerland*, are less frequently used to publish their research articles by the researchers.

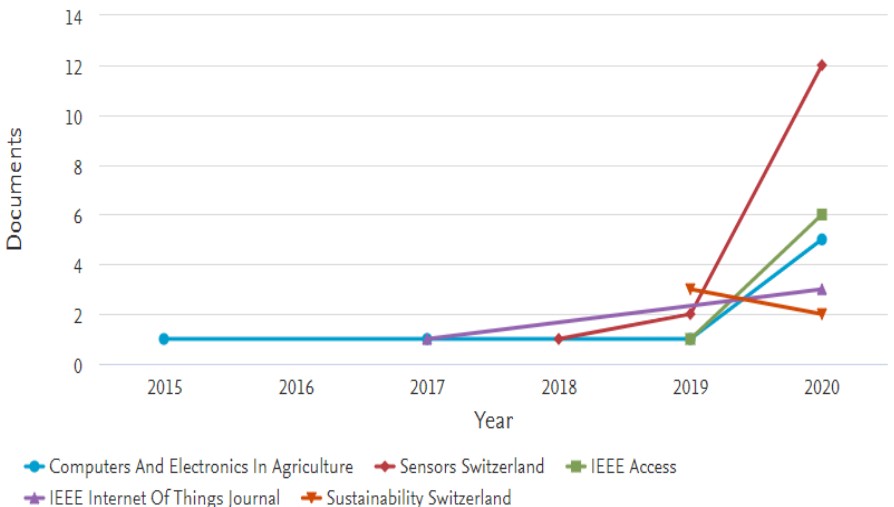

**Figure 13.** Top 5 source statistics for IoT in agriculture publications. Repository information retrieval source: http://www.scopus.com (accessed on 21 December 2020).

### 3.10. Analysis Pertaining to Funding Sponsors for IoT in Agriculture

Figure 14 provides a statistical study focused on funding sponsors in the research field of IoT in agriculture.The topmost 10 leading funding sponsors are considered and it can be clearly concluded from the statistics that the European Commission is the highest funding authority. Also out of ten, the majority of the funding sponsors' authorities are from China.

Figure 15 represents the co-occurrence of the keywords utilized in the filtered papers. It shows the way in which the keywords are interrelated with each other. The different color shows the relationship of the keywords, and the size of the dot is the importance of the keywords.

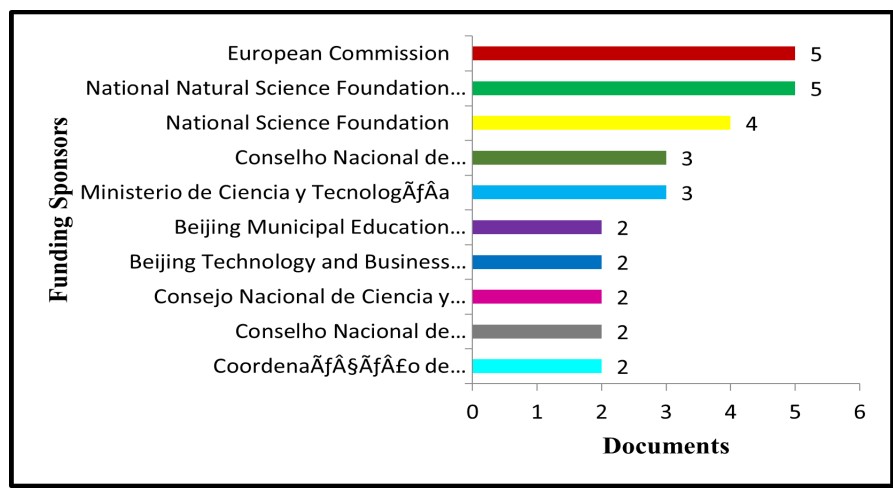

**Figure 14.** Funding Sponsors statistics for IoT in agriculture. Repository information retrieval source: http://www.scopus.com (accessed on 21 December 2020)).

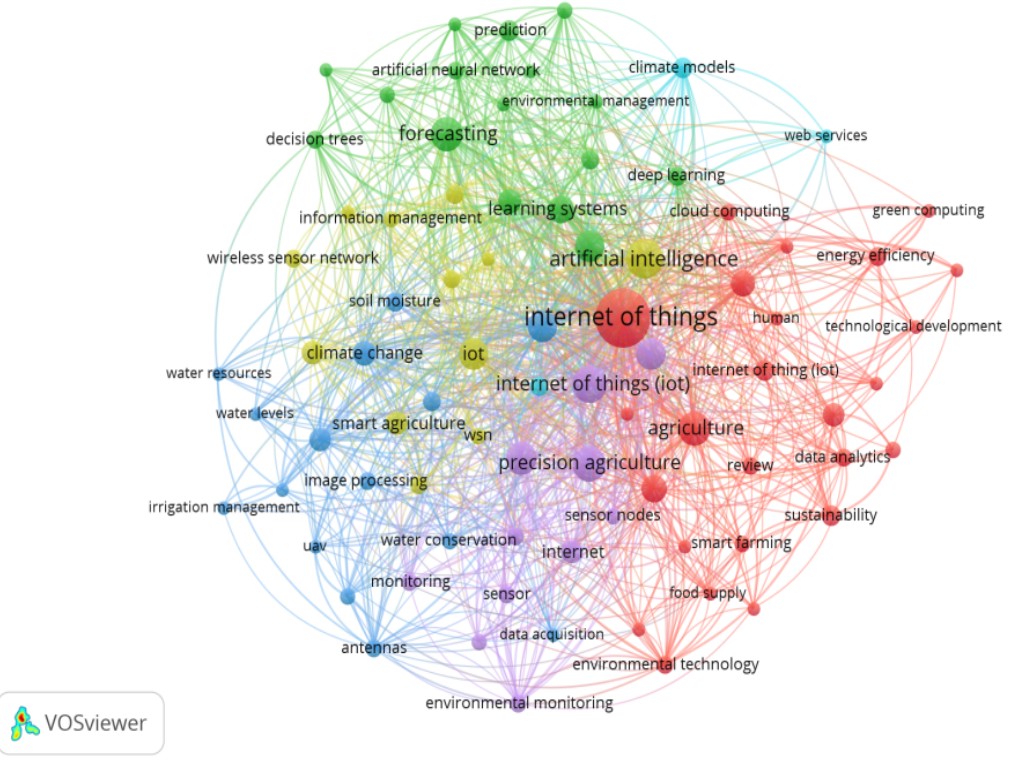

**Figure 15.** Cluster of co-occurrence of keywords. (Source: www.vosviewer.com (accessed on 21 December 2020)).

## 4. Word Cloud for IoT in Agriculture Articles

Figures 16–18 display a word cloud of three of the top five IoT in plant disease detection that are readily accessible for better analysis of the carried out bibliometric research related to IoT in plant disease detection. The aim of using a word cloud for the papers under discussion is to examine the most frequent terms, meaning that the bulk of the analysis is performed in those fields [36]. A word cloud, also known as a Tag Cloud, transforms the text data into identifiers, which are typically single words whose relative value is visualized in the cloud by their color [37]. The website www.wordclouds.com was used to build the word clouds in this post. Figure 16 shows the word cloud for article "Wireless sensor networks for agriculture: The state-of-the-art in practice and future challenges". Figure 17 shows the word cloud for the article "Deep learning for IoT big data and streaming analytics: A survey". Figure 18 shows the word cloud for the article "Big Data and cloud computing: innovation opportunities and challenges".

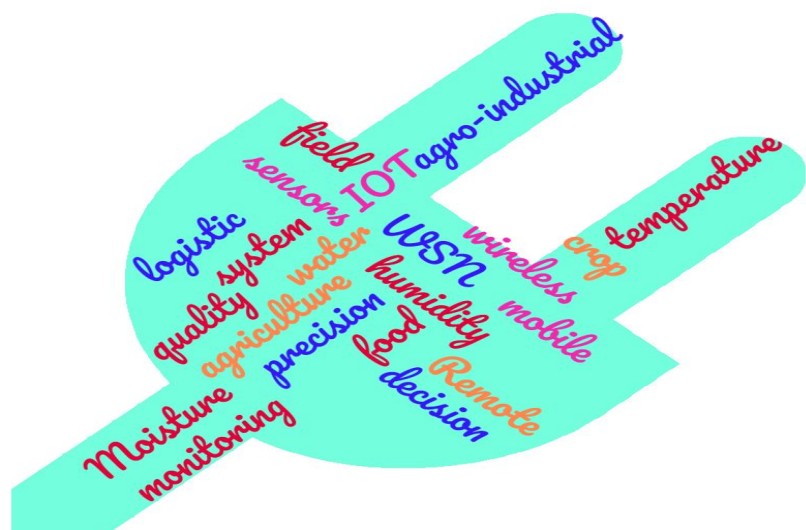

**Figure 16.** Word cloud for "Wireless sensor networks for agriculture: The state-of-the-art in practice and future challenges".

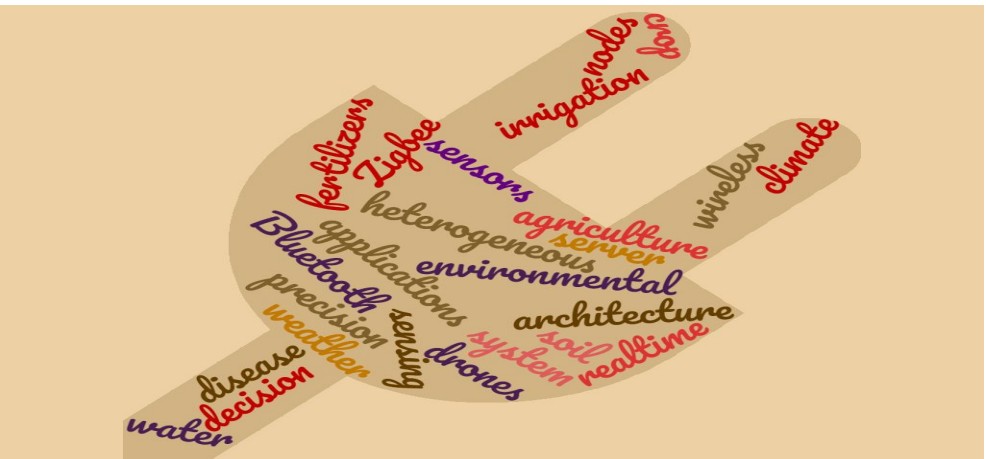

**Figure 17.** Word cloud for "Deep learning for IoT big data and streaming analytics: A survey".

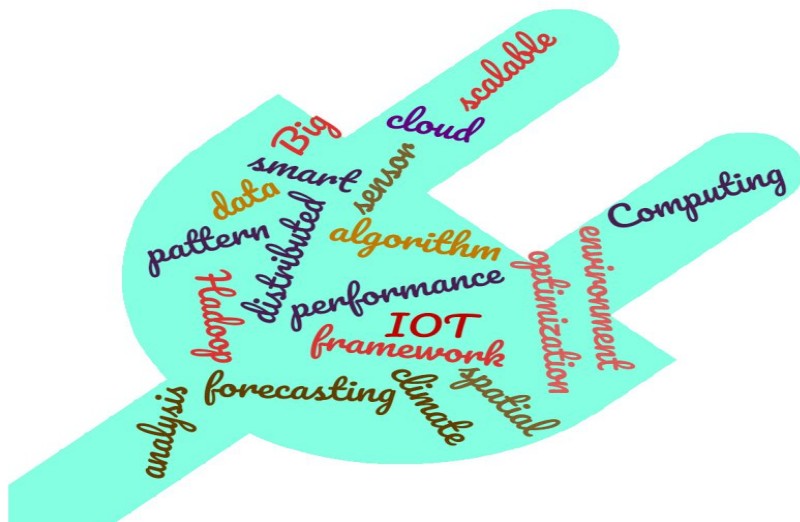

**Figure 18.** Word cloud for "Big Data and cloud computing: Innovation opportunities and challenges".

## 5. Conclusions and Future Work

A systematic assessment of the Internet of Things technologies applied in the agricultural domain for crops is yet to be available. With objective information about the current research output, it is easier to plan for necessary improvements in the infrastructure related to understanding, treating, and preventing crop diseases. Thus, it is essential to accurately assess the global and regional productivity of the ongoing research in smart agriculture.

A bibliometric analysis is used to analyze research records and scholars and generate evidence that researchers in that area can find. Writing the literature survey would be helpful. The report also assists in finding and mapping gaps in the literature. This report often contains information on emerging research developments. The author's collaboration in the work represents their interests in that area. Universities, as opposed to corporations, offer tremendous effort. From 2013 to 2020, the United States led the IoT in plant disease research. The study's results suggest that studies should be based on issues that support people who work in the field and will profit more from plants because they play such an essential role in humanity's survival.

The bulk of scholars used the English language to publish their results. It was found that very little work was placed into publishing documents in the initial years of 2013 to 2016. However, the graph is increasing every year, suggesting that 2019 and 2020 are the years in which most documents have been written in this research domain. Countries such as India and the United States are making more significant contributions to science in this field. Articles are the most common source type, accounting for 51% of the time researchers use to write their science papers. According to the findings, 36 of the 146 articles in the extracted literature are examination papers. Internet of Things, Artificial Intelligence, Precision Agriculture, Agriculture, Forecasting, Wireless Sensor Networks, Agriculture Robots, Crops, and IoT, are the prime keywords for the bibliometric study of smart agriculture. As a result, keywords are the most focused domains in smart agriculture, and we should study them further. According to the statistics, the Skolkovo Institute of Science and Technology, along with Shadrin D, has enormously contributed to this field of study. The years of 2020 and 2019 have the most significant citations, while 2016 has the lowest number of citations. Researchers should study this sector, which was previously ignored due to the need to explore the field of smart agriculture.

The current research is limited to articles contained in the Scopus dataset. The scientific records in other research databases such as Web of Science and Google Scholar are not taken into account for the current research investigation. The authors chose the search terms for the Scopus database. The researcher believes that the keyword combination selected for the study may be changed or reorganized. The findings are based on papers written

between 2013 and 2021. This research analysis takes into account records written in English. Secondary records contained in Scopus are not part of a research initiative and can be used to supplement the study.

**Author Contributions:** Conceptualization, R.R.P. and S.K.; methodology, R.R.P. and R.R.; software, R.R.P.; validation, R.R.P.; formal analysis, S.K.; resources, R.R. and R.R.P.; writing—original draft preparation, R.R.P. and S.K.; writing—review and editing, P.A., S.K.P and S.K.; supervision, S.K.; project administration, S.K.; funding acquisition, P.A. and S.K.P. All authors have read and agreed to the published version of the manuscript.

**Funding:** This research received no external funding.

**Data Availability Statement:** Not applicable.

**Conflicts of Interest:** The authors declare no conflict of interest.

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
