# Peer review of "A Bibliometric and Word Cloud Analysis on the Role of the Internet of Things in Agricultural Plant Disease Detection"

_asi, doi:10.3390/asi6010027_

Round 1

Reviewer 1 Report

1. The paper is well written and authors need to proof-read the study to make it more interesting.

2. The paper considered IoT and can consider the following papers:

 “Benchmarking health-care supply chain by implementing Industry 4.0: a fuzzy-AHP-DEMATEL approach”; “Drivers of sustainable healthcare supply chain performance: multi-criteria decision-making approach under grey environment”.

3. The paper should highlight a research framework in the introduction section to provide direction of the overall study.

Reviewer 2 Report

The manuscript "The Role of Internet of Things (IoT) in Plant Disease Detection: A Bibliometric and Word Cloud Analysis" by R.R. Patil et al. presents a bibliometric analysis of publications on the topic of IoT technology application in smart agriculture based on data  drawn from the Scopus database. The paper reports mainly about a number of bibliometric statistics of the retrieved publications such as their distribution per year, author and funding body, type (articles, reports, books, etc.) and their bibliometric impact. It follows presenting the results from a data visualization technique analysis on the words of highest frequency in a restricted number of papers (5) and a graphical representation of the frequency and connections across the set of keywords used in the examined publications.

As a general comment on the manuscript title, I would say that an analysis of the role of IoT in plant disease detection should not be restricted to smart agriculture only but encompass all the instances where such a technology is applied to plants. An appropriate use of search keywords should therefore include terms as "plants disease", "forest", "urban forestry", "plant nursery" just to name a few. Unfortunately the lack of information of the retrieved publications set - it is not listed in the references - prevents from verifying the field of the single studies although it can be to some extent hinted from fig 4.

The Authors conclude what is basically a bibliometric report with generic statements on the utility of their work ("Writing the literature survey would be useful.") and claim it assists in finding and mapping gaps in the literature and providing information on emerging research developments. Nevertheless a critical analysis of the current status and directions of the use of IoT applied to plants disease detection is not substantially carried out.

Above all, I do not see how the submitted bibliometric analysis can meet in any form the scope and aims of Applied System Innovation, which I recall hereafter:

"Applied System Innovation is  devoted to publishing research papers in the fields of integrated engineering and technology, which often combines theoretical models with numerical simulations to solve problems or make predictions that may be used to design innovative solutions, not only to be used by industry but also to affect the lives of ordinary people.

The main goal of the journal is to publish research results in applied system innovation. The ultimate aim is to discover new scientific knowledge relevant to designs of the future, to enhance technological development in a range of industries, and to improve the welfare of society."

For all the reasons above I think that the manuscript should not be considered for publication.

Reviewer 3 Report

Well-structured article. In main section there is a possibility to deep using terms and their limits.

Reviewer 4 Report

Essential Questions:

? Why was the research not extended to another database, e.g. WoS? This would allow for a broader and more objective analysis.

? Why IoT isn't a search keyword? The word "sensor" used has many uses, which could and probably caused the query to be falsified. This calls into question the credibility of e.g. Table 5.

Final conclusions ... are v.very weak.

Detailed notes:

• contradiction between the sentence in line 131 - 132 and Table 4 and Figure 1.

• why in Table 5 IoT appears twice (rows 1 and 3)?

• Inference at the beginning of the point. 3.2 is a tautology of the type: if we ask about IoT, the answer is most often about IoT.

• lots of figures de facto not leading to any conclusions,

Round 2

Reviewer 2 Report

The resubmitted version of the manuscript " The Role of Internet of Things (IoT) in Plant Disease Detection: A Bibliometric and Word Cloud Analysis" by Patil et al. has been sensibly improved regarding the writing style and the clearliness of the various sections.

The perfomed bibliographic analysis  is focused on the topic of the use of IoT in smart agriculture and this should be refelected in the title of the manuscript which is still misleading. I would suggest to change it into "" The Role of Internet of Things (IoT) in Crop Disease Detection: A Bibliometric and Word Cloud Analysis" or "" The Role of Internet of Things (IoT) in Agricultural Plants Disease Detection: A Bibliometric and Word Cloud Analysis", for instance.

That said, while it is well understandable how a bibliometric survey of this kind is a ncessary step for any scholar who is about to write an article on the above cited topic in order to decide about the possible editorial target, I still struggle in figuring out what is the scientific added value of this work in order to grant its publication on ASI. However after reckoning the improvements made, I will leave to the Editor to decide whether the manuscript's results can be of potential interest for the journal's readers and worth being published.

Author Response

-------------

The resubmitted version of the manuscript " The Role of Internet of Things (IoT) in Plant Disease Detection: A Bibliometric and Word Cloud Analysis" by Patil et al. has been sensibly improved regarding the writing style and the cleanliness of the various sections.

The performed bibliographic analysis is focused on the topic of the use of IoT in smart agriculture and this should be reflected in the title of the manuscript which is still misleading. I would suggest to change it into "" The Role of Internet of Things (IoT) in Crop Disease Detection: A Bibliometric and Word Cloud Analysis" or "" The Role of Internet of Things (IoT) in Agricultural Plants Disease Detection: A Bibliometric and Word Cloud Analysis", for instance.

Authors Response:

Thank you for your valuable comments.

The same has been changed in the manuscript.

The Role of Internet of Things (IoT) in Agricultural Plants Disease Detection: A Bibliometric and Word Cloud Analysis

That said, while it is well understandable how a bibliometric survey of this kind is a ncessary step for any scholar who is about to write an article on the above cited topic in order to decide about the possible editorial target, I still struggle in figuring out what is the scientific added value of this work in order to grant its publication on ASI. However, after reckoning the improvements made, I will leave to the Editor to decide whether the manuscript's results can be of potential interest for the journal's readers and worth being published.

Authors Response:

Thank you for your valuable comments.

Authors found many bibliometric papers in Applied System Innovation journal so that’s why targeted the same. For investigating and analysing huge quantities of scientific data, bibliometric analysis is an established and efficient technique. It allows to explore the peculiarities of a certain field's evolutionary development while highlighting its horizons. However, its use in business research is still somewhat new and frequently underdeveloped. As a result, to provide an overview of the bibliometric approach, with a special emphasis on its various methodologies, as well as step-by-step directions that can be trusted to undertake bibliometric analysis rigorously and with confidence. Lastly it also emphasises on when and how to apply bibliometric analysis in comparison to other related strategies like systematic literature reviews and meta-analysis. So please reconsider this paper.

Reviewer 4 Report

Dear Authors

Well, relying on only 1 database significantly reduces the value of the article. But let it stay that way.

With all due respect for the work done, maybe I don't understand it, but:

·       You are looking for a database of articles with the keyword "sensors", and a moment later you discuss publications according to 3 versions of the "IoT" keyword, without even commenting on the total result - where did this data come from? why no result comment?

·       in line 141/142 you write about the largest number of publications in 2018-2019, and in Table 4 year 2020 has more publications than 2018 and 2019 combined,

·       in line 153 you write that there were 74 publications in 2019 and in Figure 1 in 2020 are 74 publications,

I understand and appreciate the importance of the article, which is explicitly stressed in the revised Conclusions (… Without objective information about current research output it is difficult to plan for necessary improvements in infrastructure related to the understanding, treatment and prevention of crop diseases. … Researchers should study this sector, which was previously ignored, as a result of the need to explore the field of Smart Agriculture. … ), but in the case of ambiguities or maybe errors need to be corrected, and in the case of moving from questions to data base about "sensors", and move to answers about "IoT" simply should be clearly explained or corrected.

Round 3

Reviewer 2 Report

In view of the several revisions of the manuscript, which contributed to sensibly improve its quality, and of the confirmed editorial interest for the proposed topic, the submitted paper can now be accepted in present form.

Reviewer 4 Report

OK.